# New Male Users of Lipid-Lowering Drugs for Primary Prevention of Cardiovascular Disease: The Impact of Treatment Persistence on Morbimortality. A Longitudinal Study

**DOI:** 10.3390/ijerph17207653

**Published:** 2020-10-20

**Authors:** Isabel Aguilar-Palacio, María José Rabanaque, Lina Maldonado, Armando Chaure, José María Abad-Díez, Montse León-Latre, José Antonio Casasnovas, Sara Malo

**Affiliations:** 1Department of Preventive Medicine and Public Health, University of Zaragoza, 50009 Zaragoza, Spain; rabanake@unizar.es (M.J.R.); smalo@unizar.es (S.M.); 2Fundación Instituto de Investigación Sanitaria de Aragón (IIS Aragón), 50009 Zaragoza, Spain; mleonlatre@gmail.com (M.L.-L.); jacasas@unizar.es (J.A.C.); 3GRISSA Research Group, 50009 Zaragoza, Spain; achaure@salud.aragon.es (A.C.); jmabad@aragon.es (J.M.A.-D.); 4Department of Economic Structure, Economic History and Public Economics, University of Zaragoza, 50005 Zaragoza, Spain; lmguaje@unizar.es; 5Department of Health, Aragon Health Service (SALUD), 50017 Zaragoza, Spain; 6Biomedical Research Networking Center on Cardiovascular Diseases (CIBERCV), 28029 Madrid, Spain

**Keywords:** lipid-lowering drugs, persistence, major adverse cardiovascular event

## Abstract

The objective of this study was to analyse persistence to lipid-lowering drug use for primary prevention of cardiovascular disease (CVD) in a new users cohort, to explore all-cause and cardiovascular related morbidity, comorbidity and mortality in this group and, finally, to study the relationship between persistence and morbimortality. We selected subjects who started lipid-lowering treatment for primary prevention of CVD between 1 January 2010 and 31 December 2017 (N = 1424), and classified them as treatment-persistent or -nonpersistent. Bivariate analyses were performed to compare sociodemographic and clinical variables, morbimortality and time to event between groups. The association between morbidities was explored using comorbidity network analysis. The effect of persistence was analysed using logistic regression and Cox survival analyses. Only 38.7% of users were persistent with treatment. Persistent and nonpersistent users had similar sociodemographic and clinical profiles, although differed in age, smoking status, and glycemia. Comorbidity networks revealed that the number of co-occurring diagnoses was higher in nonpersistent than persistent users. Adjusted analyses indicated a protective effect of treatment persistence, especially against major adverse cardiovascular events (MACE), but this effect was not statistically significant. Observational studies are crucial to characterize real-world effectiveness.

## 1. Introduction

Hypercholesterolaemia is a well-known risk factor for cardiovascular disease (CVD) in people with or without a previous cardiovascular event [1]. Accordingly, CVD prevention guidelines define treatment objectives based on cholesterol levels [2]. Lipid-lowering drugs, especially statins, have been widely used for primary prevention of CVD [3]. A large body of evidence from randomized clinical trials supports the efficacy of these treatments in reducing the risk of both CVD and all-cause mortality [4,5]. However, several aspects of randomized clinical trials limit extrapolation of their findings to real-world situations. For example, only volunteer subjects are included in these trials, and patients and their circumstances are monitored much closer than typically occurs in routine practice [6]. On the other hand, real-world factors, such as contextual and behavioural patient characteristics, like real-world patterns of treatment use, contribute significantly to drug effectiveness [7] All these facts help explain the differences observed between randomized clinical trials and observational studies in terms of treatment efficacy and effectiveness. Therefore, treatment effectiveness and safety should be evaluated based on evidence obtained from observational studies [8].

Patient persistence, defined as the continuity of treatment over time, has been closely linked with the clinical benefits of treatment [9]. Many studies have described poor persistence with lipid-lowering drugs [10], especially among patients receiving long term treatment [11]. In agreement with the poor persistence rates observed among individuals prescribed lipid-lowering drugs for primary disease prevention, poor persistence with these treatments is associated with young age and prescription for primary disease prevention [7].

It is important to study the effect of patterns of treatment use on morbidity and mortality in a real environment. To this end, the objective of this study was to analyse persistence to lipid-lowering drugs for primary prevention of cardiovascular disease (CVD) in a new users cohort, to explore all-cause and cardiovascular related morbidity, comorbidity and mortality in this group and, finally, to study the relationship between persistence and morbimortality.

## 2. Materials and Methods

### 2.1. Study Design and Participants

This was a follow-up study, performed during the period 2010–2018. The population studied belong to the Aragon Workers’ Health Study (AWHS). The AWHS is a prospective longitudinal study of voluntary workers of a Spanish automobile assembly plant that was designed to evaluate the trajectories of traditional and emergent CVD risk factors and their association with the prevalence and progression of subclinical atherosclerosis in a middle-aged working population free of CVD at the beginning of the study. Recruitment of the AWHS cohort began in February 2009 and ended in December 2010. Further information on the AWHS can be found elsewhere [12].

For our analysis, the inclusion criteria were subjects who began lipid-lowering treatment (Anatomical Therapeutic Chemical [ATC] code [13] C10) between 1 January 2010 and 31 December 2017. ATC code C10 include lipid modifying agents plain (HMG CoA reductase inhibitors, fibrates, bile acid sequestransts, nicotinic acid and derivates and other lipid modifying agents) and HMG CoA reductase inhibitors in combination with other lipid modifying agents. Analyses were restricted to new users, defined as those who had not received any lipid-lowering drug prescription during the 6 months preceding the start date (N = 1601). From those selected, we excluded individuals diagnosed with any CVD before beginning lipid-lowering treatment, based on the principal diagnosis assigned during hospitalization (International Classification of Diseases, Tenth Revision [ICD-10] codes G45, G46, G81 to 83, I20 to I28, I46, I49.0, I50 and I60 to I79) (N = 113). Finally, there were only 64 women in the study population, due to the setting characteristics (automobile assembly). That is the reason we excluded them from our analysis. The final study population was 1424 men.

### 2.2. Data Sources and Variables Selected

Data were taken from several sources. From the AWHS study we obtained data on sociodemographic, work, lifestyle, and analytical variables at the beginning of the follow-up period. These data were collected by the physicians and nurses of the medical services of the factory. All study procedures were standardized. Smoking status was self-reported. Body mass index (BMI) and waist circumference were objectively recorded. For laboratory analyses, each participant provided a sample of blood and urine after overnight (>8 h) fasting. Fasting serum glucose and cholesterol were measured by spectrophotometry (ILAB 650 Chemical Analyzer, Instrumentation Laboratory SpA, Bedford, Massachusetts). Systolic blood pressure (SBP) and diastolic blood pressure (DBP)were measured 3 consecutive times using an automatic oscillometric sphygmomanometer (OMRON M10-IT; OMRON Healthcare Co. Ltd., Kyoto, Japan) with the participant sitting after a 5-min rest. Hypertension was defined as SBP > 140 mmHg or DBP > 90 mmHg. Low-density lipoprotein cholesterol was calculated using Friedewald’s formula [14]. Finally, CVD risk was assessed for each subject using the European Systematic COronary Risk Evaluation (SCORE) algorithm for low-risk CVD countries [2].

Information on lipid-lowering treatments (ATC code C10) was obtained from the Medication Consumption Information System of Aragon (Farmasalud) for the period 2010–2018. This database stores information on drugs dispensed by pharmacies for prescriptions issued via the Aragon Health System. For each subject, we obtained information on the dispensing date, the ATC code of the dispensed drug, the number of defined daily doses (DDD), and the number of packages dispensed.

End-points were all-cause hospitalization, CV events, major adverse CVD events (MACE) and mortality for the period 2010–2018. All these health results were considered for each patient after beginning lipid-lowering treatment. Data on all-cause hospitalization, CV events, and MACE, as well as the corresponding dates, were obtained from the national health system’s hospital discharge record database (CMBD). To identify CV events, we selected hospital discharges for which the principal diagnosis corresponded to a CVD code (ICD-10: G45, G46, G81 to G83, I20 to I28, I46, I49.0, I50 and I60 to I79). Data on MACE were also obtained from the principal diagnosis at hospital discharge, using ICD-10 codes I21 and I60 to I63. Finally, mortality was obtained from the National Mortality Registry.

### 2.3. Analyses

New users of lipid-lowering drugs for primary prevention of CVD were classified as persistent or nonpersistent. Persistence with this medication was evaluated between the time of treatment initiation and discontinuation during a follow-up period of 1 year. Based on the usual dosage and form of presentation of the drug, we considered that statin and fibrate prescriptions corresponded to 28 and 30 days, respectively. Subjects treated with lipid-lowering drugs was classified as persistent if they had no gap between two refills that exceeded 2.5 times the duration of the previous prescription during the 1-year follow-up period [15]. The accumulation of supplies over time was not considered. Subjects who received only one prescription were classified as nonpersistent.

We conducted a descriptive analysis to compare the sociodemographic and clinical characteristics of the two groups at baseline. Bivariate analyses (Chi-squared tests and Mann-Whitney U-test for continuous non-normal variables) were performed to examine differences between persistent and nonpersistent subjects and to evaluate differences in morbimortality and time to event according to persistence.

We next performed a comorbidity network analysis based on hospitalization diagnoses (CMBD) obtained [16]. We obtained 447 different diagnoses. The most frequent diagnoses was lipid metabolism disorders (ICD-10: E78), which appear 259 times, followed by primary hypertension (ICD-10: I10) which appear 205 times. Comorbidity networks were generated to determine which diagnoses co-occur more frequently than expected by random chance. Nodes represent diagnoses and lines are the relationship between diagnoses. The magnitude between diagnoses indicates the number of times that two diagnoses co-occur in the sample, after discarding all pairs of diagnoses that only appear once. To quantify the “strength” of the association between two diagnoses, the number of patients with a given pair of diagnoses (observed cases) was divided by the number of individuals likely to have both diagnoses by chance (expected cases). Only positive associations were considered (having one diagnosis makes it more likely to have the other). A ratio > 1 indicates an “association” between diagnoses. To avoid inclusion of associations by chance proportion difference tests were performed: the resulting p-value indicates the probability that the difference between the observed and the expected findings is not exclusively due to chance. Based on these results we were able to select statistically positive significant associations (alpha = 0.05). Finally, nodes that were not associated with any other node were discarded.

Logistic regression analyses were performed to determine the risk of hospitalization, CV events, MACE, or death as a function of treatment persistence. Crude and adjusted odds ratios (OR) and corresponding 95% confidence intervals (CI95%) were determined for nonpersistent users, taking persistent users as a reference. To examine the influence of persistence on survival, Cox regression analyses were performed and hazard ratios (HZ) and corresponding CI95% values determined. All analyses were conducted using R software 1.2.1335.

### 2.4. Ethics

This study was approved by the Aragon Research Ethics Committee (PI17/00042). Data was anonymised and participants signed informed consent. This study has not been previously conducted and current results are not overlapped with other previously published or ongoing reports.

## 3. Results

During the study period, 1424 men were newly treated with lipid-lowering drugs for primary prevention of CVD. Of these, only 551 (38.7%) were persistent with treatment. Table 1 shows descriptive statistics for both persistent and nonpersistent patients at baseline. Compared with nonpersistent patients, persistent patients were older (*p* < 0.001), accounted for greater proportion of current smokers (*p* = 0.04), had higher glycemia levels (*p* = 0.028), and had higher SCORE values (median, 2.2 vs. 2.1; *p* = 0.001).

More than one third of subjects (37.8%) were hospitalised for any cause after starting lipid-lowering treatment. 64 (4.5%) had a CV event, 26 (1.8%) a MACE. Thirty-three (33) patients died during the follow-up period. Although the frequencies of hospitalization, CV events, and MACE were higher in nonpersistent patients, no significant differences were observed between groups for these variables. Time to first hospitalization for any cause was lower in the persistent versus the nonpersistent group (median, 729 and 954 days, respectively; *p* = 0.029). Overall, the median time to CV event and to MACE was 844 and 893 days, respectively. No significant differences were observed between groups for these variables (Table 2).

The comorbidity network for new users of lipid-lowering drugs is shown in Figure 1. Red indicates diseases related to CVD or CV risk factors, such as diabetes or hypertension. All other diseases are shown in blue. Diagnoses of chronic ischemic heart disease and acute myocardial infarction co-occurred 17.6 times more often than expected by chance, and were also associated with long-term drug therapy. Among the other diseases recorded, the strongest association was observed for hypertensive chronic kidney disease and chronic kidney disease (which co-occurred 75.7 times more often than expected by chance). 

Figure 2 shows the comorbidity networks for persistent and nonpersistent new users of lipid-lowering drugs for primary CVD prevention. The number of co-occurring diagnoses was higher for nonpersistent than persistent users. Among nonpersistent users there was a high probability of comorbid chronic kidney disease and hypertensive kidney disease (these diagnoses co-occurred 51.9 times more often than expected by chance) and comorbid chronic ischemic heart disease and acute myocardial infarction (co-occurred 15.4 times more often than expected by chance). We also observed a significant association between hypertension and lipid metabolism disorders. By contrast, among persistent users the only diseases for which a significant association was observed were hypertension and lipid metabolism disorders (co-occurred 1.6 times more often than expected by chance).

Comorbidity networks stratified according to SCORE values are provided in Appendix A. Table 3 and Figure 3 show the results of the multivariate analyses. Table 3 shows the risk of all-cause hospitalization, CVD events, MACE, and death among nonpersistent users, compared with persistent users and adjusted by SCORE value. After adjusting by SCORE, nonpersistent users had a higher risk of all-cause hospitalization, CVD events, and MACE than persistent users. Although none of these differences were statistically significant, the difference between groups in the risk of all-cause hospitalization approached significance (*p* = 0.060). Higher SCORE values were associated with a higher probability of hospitalization, CVD events, MACE, and mortality.

Results shown in Figure 3 were similar to those in Table 3. The highest risk during the follow-up for the nonpersistent group, compared with persistent users, was observed for MACE (HR, 1.5; CI95%, 0.6–3.6), although this difference was not significant.

## 4. Discussion

In our population of new users of lipid-lowering drugs for primary prevention of CVD, only 38.7% were persistent with treatment. Persistent and nonpersistent subjects had a similar profile, although we observed significant differences between groups regarding age, smoking status, and glycemia, and a slightly higher SCORE value in persistent users. The frequency of all-cause hospitalization, CV events, MACE, and death was similar for persistent and nonpersistent users. Regarding the time to hospitalization, it was significantly higher in non-persistent. As we are considering any cause of hospitalization, we cannot associate it to treatment. Nonetheless, comorbidity network analysis revealed a higher number of co-occurred diagnoses in nonpersistent users than persistent users. We observed a strong association between CVD and CV risk factors such as hypertension or diabetes. Finally, the adjusted regression analyses revealed a protective effect of treatment persistence for MACE, although this effect was not statistically significant. A higher SCORE value was a risk factor for all the results analysed.

As we reported in a previous study [17], the prevalence of CVD risk factors in the AWHS cohort is high. We observed a similarly high prevalence in the new users of lipid-lowering drugs included in our study. Persistence with lipid-lowering drugs for primary prevention of CVD was low, albeit similar to that reported in other observational studies [18]. This is an important public health problem, as persistence with lipid-lowering drugs is associated with reductions in cardiovascular events and mortality [18]. In this sense, real-world studies provide a good evidence to reflect the true persistence with pharmacological treatments. Multiple factors appear to influence persistence with these drugs, including sociodemographic characteristics, medical history, and healthcare utilization [19]. A previous study of the persistence with lipid-lowering drugs in the AWHS cohort [7] identified the low age of the population as a key contributing factor to poor persistence. In the present study, there were differences between persistent and nonpersistent users in age, which was also higher in persistent users, and SCORE value, which was significantly higher in the persistent group. Other study conducted in a population of newly treated middle-aged individuals with dyslipidaemia reported that subjects with other CV risk factors such as diabetes or hypertension were the most likely to persist with statin therapy [20]. This finding could help explain why individuals with a high risk of CVD are more persistent to treatment.

To the best of our knowledge, this is the first study to perform a comorbidity network analysis of patients treated with lipid-lowering drugs. This analytical approach provides us with a broader perspective of the clinical profile of individual patients, who may have multiple comorbid diseases or conditions. Comorbidity network analysis is a novel approach to the study of differences in morbidity between groups (in this case persistent versus nonpersistent patients) [21,22]. Our comorbidity networks revealed an association between dyslipidaemia and hypertension for the entire study population (i.e., for both persistent and nonpersistent users). This association is well documented in the literature [23]. We also observed several interesting associations for illnesses other than CVD. CV risk factors have been associated with airflow obstruction [24], and this association was observed for the entire group studied and for the nonpersistent group. A study conducted in the UK reported high rates of CVD among individuals who also had malignancies [21]. The researchers found that some of these co-occurrences could be attributed to lifestyle factors. Although we observed no association between malignancies and CVD in our study population, malignancies were more frequent than can be explained by chance in the entire dyslipidaemia cohort, supporting this hypothesis. Finally, we observed differences in comorbidity between persistent and nonpersistent subjects. Comorbidities were more prevalent in nonpersistent patients. This fact could be explained as a consequence of non-persistence, but also as a cause, because patients with a high number of diseases tend to reduce drugs to the minimum necessary number. The strong association observed between chronic kidney disease and hypertensive kidney disease in non-persistent deserves further research, as it could be related to adverse effects or to other coprescriptions, which could be also the reason of non-persistence. The most important difference between groups related to concurrent diagnoses of chronic ischemic heart disease and acute myocardial infarction; this combination of diagnoses was only detected in nonpersistent patients, and may be explained by the poorer management of cholesterol levels in this group. Nonetheless, this result should be interpreted cautiously.

Although the crude and the adjusted regression analyses revealed a protective effect of treatment persistence against all-cause hospitalization, CV events, and MACE, this effect was nonsignificant in all cases. The failure of drugs to reduce all-cause mortality in primary prevention contexts has been associated with the low risk of death in the corresponding user populations [25]. Supporting this hypothesis, the median SCORE value for our entire study population was 2.1, indicating a low risk of death by CVD. Despite these low SCORE values, we found that higher SCORE values were associated with a greater risk of all-cause hospitalization, CV events, MACE, and death, validating our results.

Although many randomized clinical trials have demonstrated an association between lipid-lowering drug use and reductions in CVD [25], observational studies investigating this association in primary prevention and in low-risk patients are scarce. In one such observational study comparing current users vs. never users, Danaei et al. [26] reported potential benefit effects of statin therapy for primary prevention of coronary heart disease.

In the present study, we compared persistent versus nonpersistent subjects. This approach may have diminished the protective effect observed when comparing users and never users. Using real-life data Shalev et al. [27] evaluated the effect of persistence with statin therapy on low-density lipoprotein cholesterol (LDL) levels. The authors observed a strong association between persistence and decreased LDL levels in both primary and secondary prevention contexts, and proposed that this effect was in agreement with the efficacy of statins reported in randomized clinical trials. Finally, a meta-analysis of observational studies analysing statin effectiveness in primary prevention [28] found that the most adherent users had 22% fewer CVD events than the least adherent. This effect was even stronger in our study population, in which nonpersistent users had a 38% higher risk of CVD than persistent users, although our results were not statistically significant.

Our study has some limitations, mainly related to the data sources used. New users were classified as persistent or nonpersistent based on information available in the Farmasalud database. Farmasalud collects information on drugs dispensed through the public healthcare system. Although some dispensations may not be included, it is thought to cover approximately 98.5% of the population. Some information is not available in Farmasalud, such as indication or treatment regimen, so we could not estimate treatment intensity. Also, our analysis is based on the assumption that drug purchasing equates to drug consumption, which is not necessarily true. Nonetheless, pharmacy records provide an easy and affordable way to estimate persistence in a real-world context [29]. Also, we do not know the causes of no persistence, so they could be related to adverse events or a medical decision. Information related with behaviours, such as diet or exercise, or genetics, was not available for all the subjects included. That is the reason why we could not consider it. Limitations associated with the use of comorbidity networks should also be considered. We selected all diagnoses (principal and secondary) at hospital discharge after beginning lipid-lowering treatment. Nonetheless, some of these diseases may have been established before users began treatment, as sometimes secondary diagnoses are used to collect relevant clinical antecedents. Because we cannot rule out the possibility that some of these conditions were already present when the patient began treatment, the observed association between treatment use and comorbidity should be interpreted with caution. Other limitations include the small sample size of the nonpersistent group, which complicated detection of statistically significant associations between diagnoses in this group. The lack of significant differences may also be associated with the young age of the new lipid-lowering drug users (median age, 52 years). Also, the low number of CV events identified and the median duration of follow-up (2136 days; Percentile 25, 1498; Percentile 75, 2505) was insufficient to detect statistically significant differences between groups. There were only 33 deaths in the cohort during the follow-up. The most frequent cause of death was lung cancer (9 deaths). Only 7 people died because of CVD in our study population, 4 in the persistent group and 3 in the nonpersistent group. For this reason, we could not analyse deaths by CVD.

This study also has several important strengths. First, by availing of data collected for the AWHS cohort we were able to analyse data from different sources (clinical and administrative), providing a broader perspective on CVD primary prevention. Second, we used hospital discharge records (the CMBD database) to identify CVD and other diagnoses. This data source has proven to be a useful tool to provide quality information in the field of CVD [30,31]. Third, our analysis was limited to incident users of lipid-lowering treatment, thereby diminishing differences between our results and those of randomized clinical trials [8]. Finally, although observational studies are affected by certain biases, they overcome several of the limitations associated with randomized clinical trials, including the short observation time and the lack of external validity [32].

## 5. Conclusions

Treatment persistence was low in our population of new users of lipid-lowering drugs. We observed a high number of comorbid conditions coinciding with CV diseases, particularly among nonpersistent patients. This morbidity profile should be considered in order to improve patient management. Although treatment persistence had a protective effect, especially against the development of MACE, these results were not statistically significant. Further observational studies will be necessary to evaluate the real-world effectiveness of lipid-lowering drugs while taking into account patterns of use.

## Figures and Tables

**Figure 1 ijerph-17-07653-f001:**
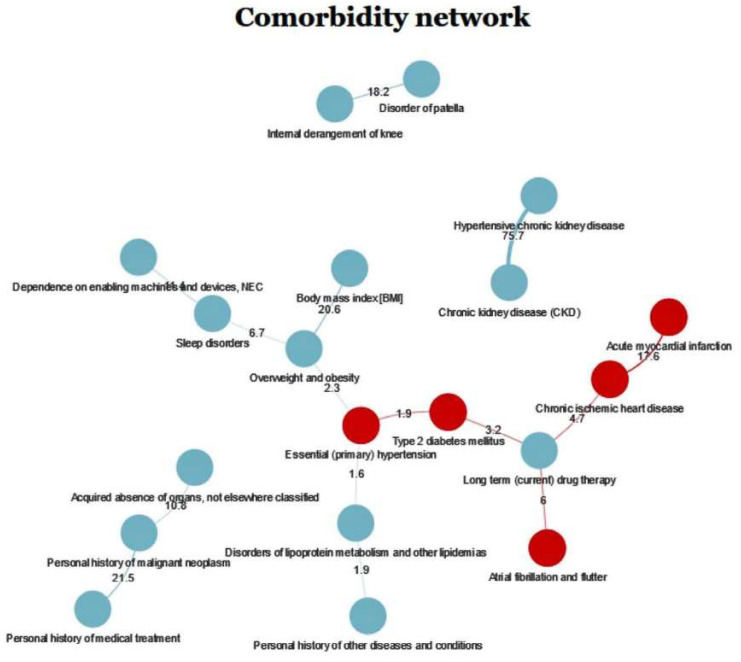
Comorbidity network for new male users of lipid-lowering drugs. Nodes represent diagnoses (in red: cardiovascular diseases or cardiovascular risk factors). Lines are the relationship between diagnoses. The magnitude between diagnoses indicates the number of times that two diagnoses co-occur in the sample, after discarding all pairs of diagnoses that only appear once.

**Figure 2 ijerph-17-07653-f002:**
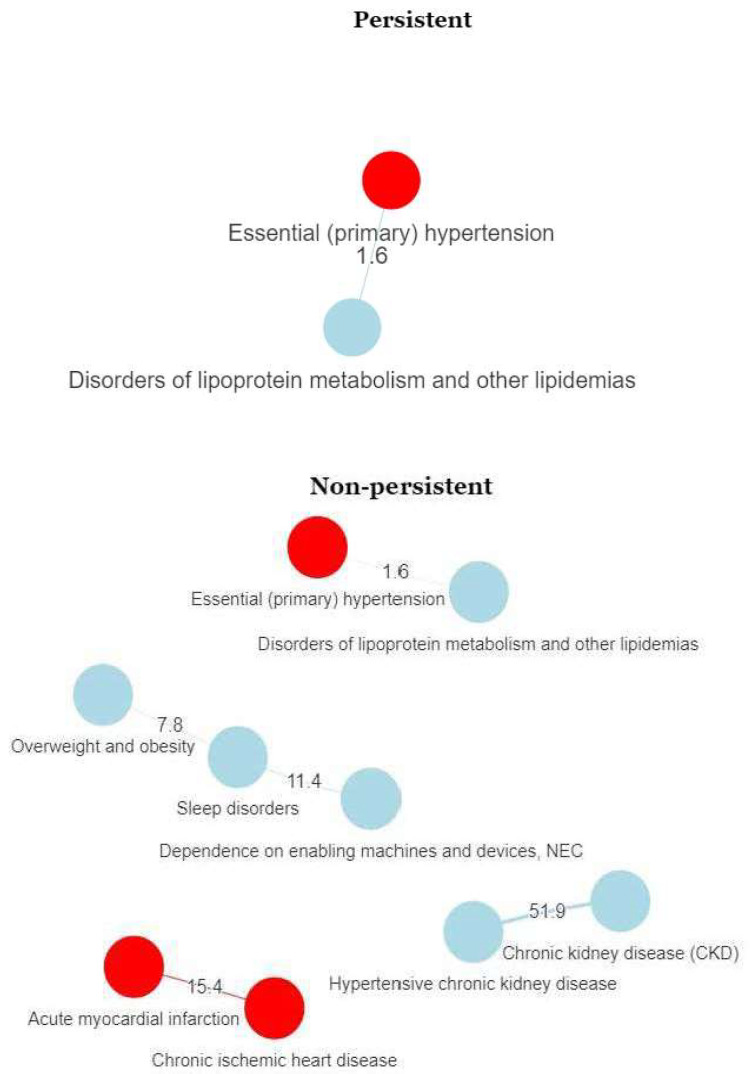
Comorbidity network for new male users of lipid-lowering drugs: persistent versus nonpersistent users. Nodes represent diagnoses (in red: cardiovascular diseases or cardiovascular risk factors). Lines are the relationship between diagnoses. The magnitude between diagnoses indicates the number of times that two diagnoses co-occur in the sample, after discarding all pairs of diagnoses that only appear once.

**Figure 3 ijerph-17-07653-f003:**
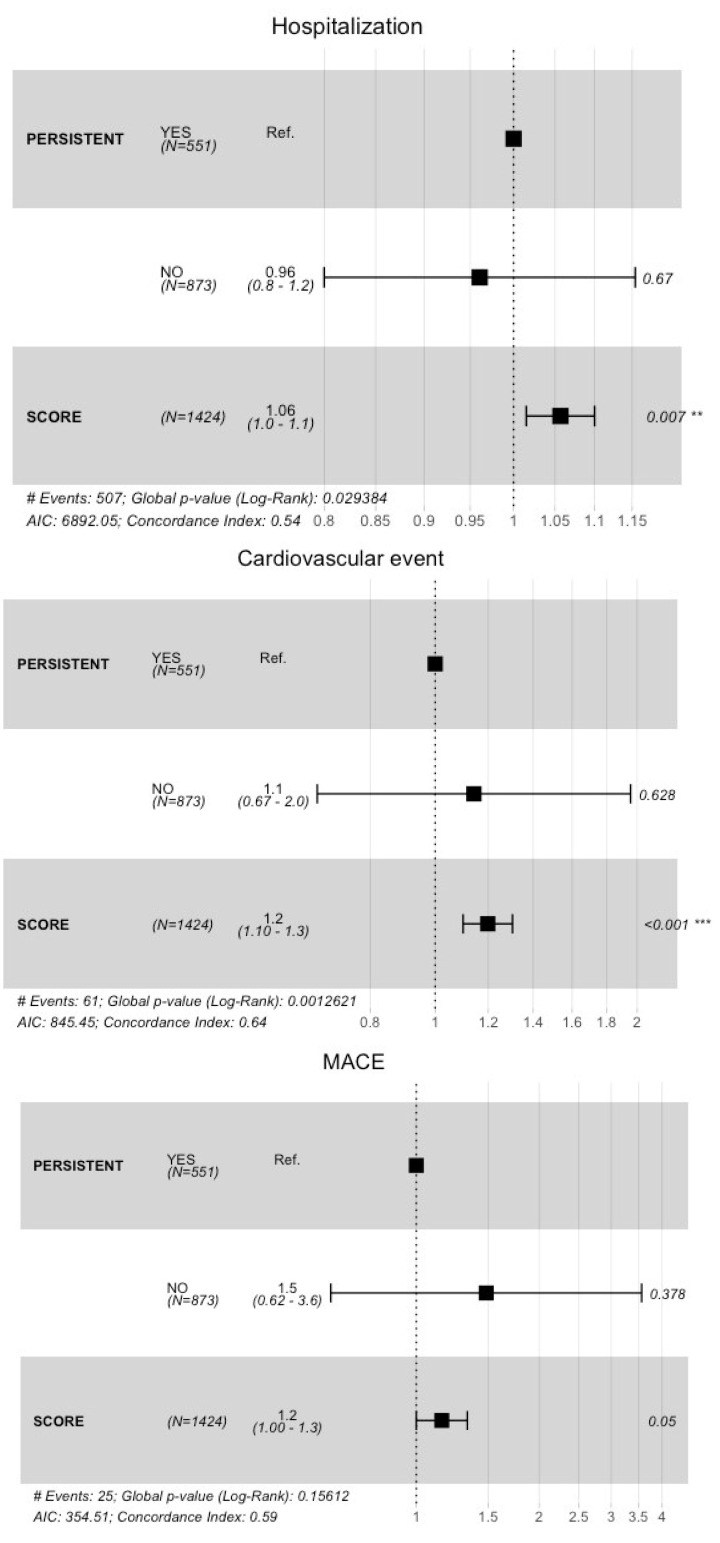
Persistence with lipid-lowering treatment and time to hospitalization, cardiovascular event, major adverse cardiovascular event (MACE), and death in men. Cox regression analyses adjusted by SCORE value.

**Table 1 ijerph-17-07653-t001:** Descriptive statistics for new male users of lipid-lowering drugs for primary cardiovascular disease prevention: overall and stratified by persistence.

	Overall (N = 1424)	Nonpersistent (N = 873)	Persistent (N = 551)	*p*
Age				<0.001 *
<50	454 (31.9%)	303 (34.7%)	151 (27.4%)	
50–55	649 (45.6%)	412 (47.2%)	237 (43.0%)	
>55	321 (22.5%)	158 (18.1%)	163 (29.6%)	
Manual work	1215 (86.0%)	734 (84.9%)	481 (87.8%)	0.116
Work shift				0.691
Rotating morning-evening	811 (57.4%)	491 (56.8%)	320 (58.4%)	
Rotating morning-evening-night	290 (20.5%)	184 (21.3%)	106 (19.3%)	
Central	124 (8.78%)	79 (9.1%)	45 (8.2%)	
Night	188 (13.3%)	111 (12.8%)	77 (14.1%)	
Smoking status				0.040 *
Never	342 (25.1%)	198 (23.6%)	144 (27.4%)	
Former	537 (39.3%)	352 (42.0%)	185 (35.2%)	
Current	486 (35.6%)	289 (34.4%)	197 (37.5%)	
BMI				0.715
<25	193 (14.0%)	121 (14.3%)	72 (13.5%)	
≥25	1185 (86.0%)	723 (85.7%)	462 (86.5%)	
Hypertension	886 (64.9%)	560 (66.7%)	326 (61.9%)	0.075
Total cholesterol mg/dL				0.682
<200	225 (15.8%)	141 (16.2%)	84 (15.2%)	
≥200	1196 (84.2%)	729 (83.8%)	467 (84.8%)	
LDL cholesterol mg/dL				0.504
<100	97 (7.23%)	63 (7.66%)	34 (6.54%)	
≥100	1245 (92.8%)	759 (92.3%)	486 (93.5%)	
Glycemia mg/dL, median [P25; P75]	98.0 [89.0; 108.0]	97.0 [89.0; 107.0]	99.0 [90.0; 110.0]	0.028 *
SCORE, median [P25; P75]	2.1 [1.3; 3.4]	2.1 [1.2; 3.2]	2.2 [1.4; 3.6]	0.001 *

N, number; *p*, *p*-value for Chi-squared and Mann-Whitney U-test (glycemia and SCORE value); BMI, body mass index; LDL, low-density lipoprotein; SCORE, European Systematic Coronary Risk Evaluation value for low CVD countries; * *p* < 0.05.

**Table 2 ijerph-17-07653-t002:** Morbimortality and time to event in new male users of lipid-lowering drugs for primary cardiovascular disease prevention: overall and stratified by persistence.

	Overall (N = 1424)	Nonpersistent (N = 873)	Persistent (N = 551)	*p*
All-cause hospitalization	538 (37.8%)	342 (39.2%)	196 (35.6%)	0.190
Cardiovascular event	64 (4.5%)	43 (4.9%)	21 (3.8%)	0.391
Major adverse cardiovascular event	26 (1.8%)	18 (2.1%)	8 (1.5%)	0.526
Death	33 (2.3%)	19 (2.2%)	14 (2.5%)	0.791

Time to hospitalization (days)	844 [390; 1462]	954 [384; 1558]	729 [394; 1183]	0.029 *
Time to cardiovascular event (days)	844 [399; 1385]	985 [530; 1402]	635 [394; 1295]	0.320
Time to major adverse cardiovascular event (days)	893 [420; 1281]	944 [521; 1281]	574 [264; 1024]	0.222
Time to death (days)	666 [395; 1344]	659 [207; 1414]	966 [618; 1254]	0.362

N, number; *p*, *p*-value for Chi-squared and Mann-Whitney U-test (time variables); MACE, major adverse cardiovascular event; * *p* < 0.05. Time variables are expressed as the median [P25; P75].

**Table 3 ijerph-17-07653-t003:** Persistence with lipid-lowering treatment and risk of hospitalization, cardiovascular event, major adverse cardiovascular event (MACE), and death in men. Logistic regression analyses (crude and adjusted by SCORE value).

	Hospitalization				Cardiovascular Event				Major Adverse Cardiovascular Event				Death			
	Crude OR (CI95%)	*p*	Adjusted OR (CI95%)	*p*	Crude OR (CI95%)	*p*	Adjusted OR (CI95%)	*p*	Crude OR (CI95%)	*p*	Adjusted OR (CI95%)	*p*	Crude OR (CI95%)	*p*	Adjusted OR (CI95%)	*p*
Persistent	Ref.		Ref.		Ref.		Ref.		Ref.		Ref.		Ref.		Ref.	
Non-persistent	1.17 (0.94–1.46)	0.172	1.25 (0.99–1.57)	0.060	1.31 (0.78–2.27)	0.324	1.38 (0.81–2.44)	0.251	1.43 (0.64–3.50)	0.405	1.70 (0.73–4.40)	0.241	0.85 (0.43–1.75)	0.656	0.90 (0.45–1.86)	0.777
SCORE			1.07 (1.01–1.13)	0.017 *			1.21 (1.10–1.32)	<0.001 *			1.15 (0.98–1.32)	0.055			1.24 (1.10–1.38)	<0.001 *

OR, odds ratios; CI95%, 95% confidence interval; Ref., reference category; SCORE, European Systematic Coronary Risk Evaluation value for low CVD countries; * *p* < 0.05. Crude OR, OR for persistence; Adjusted OR, adjusted by SCORE value.

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
