# Peer review of "New Male Users of Lipid-Lowering Drugs for Primary Prevention of Cardiovascular Disease: The Impact of Treatment Persistence on Morbimortality. A Longitudinal Study"

_ijerph, 2020, doi:10.3390/ijerph17207653_

Round 1
Reviewer 1 Report
The authors studied the effect of treatment persistent of lipid-lowering medication on primary prevention of cardiovascular diseases and overall mortality in an observational study. The research question is relevant, the study design is appropriate, given the fact they aim to study effectiveness of the treatment, and the statistical analyses are appropriate. Some interpretations derived from the results as well as their discussion could be improved.
Title: study design is missing
Abstract:
Lines 25 and 29 contain repeated information. Please specify.
First you write: “ Adjusted analyses indicated a protective effect of treatment persistence, especially against major 27 adverse cardiovascular events (MACE), although it was not statistically significant.”, and then “Persistence with lipid-lowering drugs has a protective effect, especially against MACE.”. Please explain better.
Introduction
Methods
Line 75: Why were so few women included initially?
Line 75 It should be stated in the title that only men were included.
Line 99: how was MACE defined?
Line 111: The word “worker” seems inappropriate, please be more specific
Discussion
Line221: Are there other studies showing a better adherence to lipid lowering therapies? Why do you consider it a low adherence? Compared to what?
Line 228: If the effect is not significant, you should not emphasize it with “particularly….”
Line 237: Has any other paper reported that younger people tend to be less adherent to lipid lowering therapy? Or to medical therapy in general?
Line 263: This is a misleading interpretation. The way you write this sentence, the reader concludes it is significant.
Conclusion:
Here again you overinterpret a non-significant association.
Overall language
It is highly recommended that a native English speaker corrects language styling and punctuation.
Figures. Please provide a more detailed explanation on how to interpret these images in the figure legends.
Author Response
On behalf of all the co-authors I would like to thank the reviewer for his/her valuable comments, which have been very useful to improve the quality of our manuscript. Please see our point-by-point response (cursive) to the comments below.
The authors studied the effect of treatment persistent of lipid-lowering medication on primary prevention of cardiovascular diseases and overall mortality in an observational study. The research question is relevant, the study design is appropriate, given the fact they aim to study effectiveness of the treatment, and the statistical analyses are appropriate. Some interpretations derived from the results as well as their discussion could be improved.
Title: study design is missing.
The title has been modified to include this information. The current title is: “New male users of lipid lowering drugs for primary prevention of cardiovascular disease: the impact of treatment persistence on morbimortality in men. A longitudinal study”.
Lines 25 and 29 contain repeated information. Please specify.
Line 29 has been deleted.
First you write: “Adjusted analyses indicated a protective effect of treatment persistence, especially against major 27 adverse cardiovascular events (MACE), although it was not statistically significant.”, and then “Persistence with lipid-lowering drugs has a protective effect, especially against MACE.”. Please explain better.
Abstract has been modified to avoid redundant information.
Line 75: Why were so few women included initially?
The low number of women is due to the characteristics of the study setting (an automobile assembly). We have indicated it in the text as follows:
“Finally, there were only 64 women in the study population, due to the setting characteristics (automobile assembly). That is the reason we excluded them from our analysis. The final study population was 1,424 men”.
Line 75 It should be stated in the title that only men were included.
As suggested by the reviewer, it has been stated.
Line 99: how was MACE defined?
MACE is defined as “major adverse cardiovascular event”. This is a frequently used health result in cardiovascular epidemiology. MACE includes only some specific diagnose and its definition can differ. In this case, ICD-10 codes included were I21 and I60-I63 (myocardial infarction, stroke and nontraumatic subarachnoid hemorrhage).
Line 111: The word “worker” seems inappropriate, please be more specific.
This term has been changed by “Subjects treated with lipid-lowering drugs”.
Line221: Are there other studies showing a better adherence to lipid lowering therapies? Why do you consider it a low adherence? Compared to what?
Patient persistence is defined as the continuity of treatment over time. As we point out in the discussion, only 38.7% of the patients were persistent to treatment. We consider that this percentage is far away of the results needed to get a proper CVD management. Nonetheless, this is similar to other real-world studies. In this sense, we have added a new reference to put our results into a global context:
Deshpande S, Quek RGW, Forbes CA, et al. A systematic review to assess adherence and persistence with statins. Curr Med Res Opin. 2017. doi:10.1080/03007995.2017.1281109
Line 228: If the effect is not significant, you should not emphasize it with “particularly….”
We have deleted this expression.
Line 237: Has any other paper reported that younger people tend to be less adherent to lipid lowering therapy? Or to medical therapy in general?
Thank you for this comment. This fact has already been described in literature:
Perreault, S., Blais, L., Dragomir, A., Bouchard, M. H., Lalonde, L., Laurier, C., & Collin, J. (2005). Persistence and determinants of statin therapy among middle-aged patients free of cardiovascular disease. European journal of clinical pharmacology, 61(9), 667-674.
Vinogradova, Y., Coupland, C., Brindle, P., & Hippisley-Cox, J. (2016). Discontinuation and restarting in patients on statin treatment: prospective open cohort study using a primary care database. bmj, 353, i3305.
Old people tend to be more persistent to CVD preventive treatment. This fact has been associated with a higher prevalence of CVD risk factors, that make patients be aware of their own disease. Nonetheless, this relationship cannot be applied to ancient people, when other factors like medication complexity, dementia or adverse events interact.
Line 263: This is a misleading interpretation. The way you write this sentence, the reader concludes it is significant.
This sentence has been rewritten as follows: “Although the crude and the adjusted regression analyses revealed a protective effect of treatment persistence against all-cause hospitalization, CV events, and MACE, this effect was nonsignificant in all cases”.
Here again you overinterpret a non-significant association.
We have rewritten this sentence to avoid misinterpretation.
Overall language: It is highly recommended that a native English speaker corrects language styling and punctuation.
The manuscript has been reviewed by a native speaker.
Figures. Please provide a more detailed explanation on how to interpret these images in the figure legends.
Explanation has been improved as follows:
“Figure 1. Comorbidity network for new users of lipid-lowering drugs. Nodes represent diagnoses (in red: cardiovascular diseases or cardiovascular risk factors). Lines are the relationship between diagnoses. The magnitude between diagnoses indicates the number of times that two diagnoses co-occur in the sample, after discarding all pairs of diagnoses that only appear once”.

Reviewer 2 Report
Dear Authors,
thank you for allowing me to revise your original paper entitled: NEW USERS OF LIPID-LOWERING DRUGS FOR PRIMARY PREVENTION OF CARDIOVASCULAR DISEASE: THE IMPACT OF TREATMENT PERSISTENCE ON MORBIMORTALITY.
The manuscript is interesting but needs a thorough rewriting to deliver a more straightforward message. The general methods are amenable to criticisms; however, the authors reasonably clearly disclosed their limitations since there is no proof that buying the drug means taking it correctly. However, it is surely a smart but rough way to assess the adherence to the treatment, and many of the observations of the authors deserve to be highlighted and discussed.
The low adherence to treatment of new-users of the statins is an important message of this real-world study that deserves further discussion in the discussion section as a possible cause of distance between the ideal life of the RCT and the real-world of the poli-comorbidities, poli-pharmacotherapies of the elderly patients. The number of lives that could be saved with better adherence represents a major problem of public health.
The higher incidence of complications in the group of patients with low persistence could be a consequence but also a possible cause of low persistence since, commonly, patients with many comorbidities and AE tend to reduce the number of drugs to the strictly necessary. Indeed, please add a sentence to highlight this dual effect of low-persistence.
Unfortunately, the study doesn't report the indication to statin as probably the persistence is higher in patients having a more mandatory indication (i.e. secondary prevention or recent AMI) that are patients more prone to follow strictly the indications of the clinicians' respect to younger patients undergoing primary prevention. This potential flawing factor should be addressed and discussed. Please provide pre-treatment parameters that could permit us to understand the indication of statins: Previous AMI, Recent AMI, Primary prevention, hyperlipidemia, diabetes, etc.
No data are reported regarding the need to stop the treatment due to Adverse Events: how many had an increase of CPK, transaminases, or other justifiable cause to stop the treatment.
How do the authors explain the lower mortality (numerical) of the non-persistent and the significantly higher time to hospitalization and MACE? Please answer and add also a sentence in the discussion.
The comorbidity network represents one of the most engaging results of this study since the interrelation between risk factors represents one of the most intriguing and underinvestigated topics but unfortunately, Figures 1, 2, and 3 are not correctly uploaded and should be resubmitted.
The number of co-occurring diagnoses was higher for nonpersistent than persistent users. Among nonpersistent users, there was a high probability of comorbid chronic kidney disease and hypertensive kidney disease. Do the authors think the non-persistence could be due to AE or to the excessive load of drugs (number of active drugs prescribed) in these patients?
Best regards.
Author Response
On behalf of all the co-authors I would like to thank the reviewer for his/her valuable comments, which have been very useful to improve the quality of our manuscript. Please see our point-by-point response (cursive) to the comments below.
Thank you for allowing me to revise your original paper entitled: NEW USERS OF LIPID-LOWERING DRUGS FOR PRIMARY PREVENTION OF CARDIOVASCULAR DISEASE: THE IMPACT OF TREATMENT PERSISTENCE ON MORBIMORTALITY.
The manuscript is interesting but needs a thorough rewriting to deliver a more straightforward message. The general methods are amenable to criticisms; however, the authors reasonably clearly disclosed their limitations since there is no proof that buying the drug means taking it correctly. However, it is surely a smart but rough way to assess the adherence to the treatment, and many of the observations of the authors deserve to be highlighted and discussed.
Thank you very much for your comments. The abstract has been reduced in order to improve it and also some changes have been made to the introduction, discussion and conclusions.
Regarding the fact that buying the drug does not mean that the patient take it, as it has been stated in the discussion, this is a common limitation in real-world studies. Pharmacy records are a common database used in pharmacoepidemiology. Although it is true that they only capture prescriptions that are actually dispensed, they provide an easy and affordable way to estimate adherence and persistence to medication. To clarify this fact, we have added this explanation and a new reference to the limits section:
“Also, our analysis is based on the assumption that drug purchasing equates to drug consumption, which is not necessarily true. Nonetheless, pharmacy records provide an easy and affordable way to estimate persistence in a real-world context29.”
29 Kardas P, Aguilar-Palacio I, Almada M, et al. The Need to Develop Standard Measures of Patient Adherence for Big Data: Viewpoint. J Med Internet Res. 2020;22(8):e18150. Published 2020 Aug 27. doi:10.2196/18150.
The low adherence to treatment of new-users of the statins is an important message of this real-world study that deserves further discussion in the discussion section as a possible cause of distance between the ideal life of the RCT and the real-world of the poli-comorbidities, poli-pharmacotherapies of the elderly patients. The number of lives that could be saved with better adherence represents a major problem of public health.
Thank you for this suggestion. We have gone into detail in this idea in the discussion section. This is the final paragraph:
“Persistence with lipid-lowering drugs for primary prevention of CVD was low, albeit similar to that reported in other observational studies18. This is an important public health problem, as persistence with lipid-lowering drugs is associated with reductions in cardiovascular events and mortality18. In this sense, real-world studies provide a good evidence to reflect the true persistence with pharmacological treatments. Multiple factors appear to influence persistence with these drugs, including sociodemographic characteristics, medical history, and healthcare utilization19. A previous study of the persistence with lipid-lowering drugs in the AWHS cohort8 identified the low age of the population as a key contributing factor to poor persistence. In the present study, there were differences between persistent and nonpersistent users in age, which was also higher in persistent users, and SCORE value, which was significantly higher in the persistent group. Other study conducted in a population of newly treated middle-aged individuals with dyslipidaemia reported that subjects with other CV risk factors such as diabetes or hypertension were the most likely to persist with statin therapy20. This finding could help explain why individuals with a high risk of CVD are more persistent to treatment”.
The higher incidence of complications in the group of patients with low persistence could be a consequence but also a possible cause of low persistence since, commonly, patients with many comorbidities and AE tend to reduce the number of drugs to the strictly necessary. Indeed, please add a sentence to highlight this dual effect of low-persistence.
We have added this idea to the discussion section. The new paragraph is as follows:
“Finally, we observed differences in comorbidity between persistent and nonpersistent subjects. Comorbidities were more prevalent in nonpersistent patients. This fact could be explained as a consequence of nonpersistence, but also as a cause, because patients with a high number of diseases tend to reduce drugs to the minimum necessary number”.
Unfortunately, the study doesn't report the indication to statin as probably the persistence is higher in patients having a more mandatory indication (i.e. secondary prevention or recent AMI) that are patients more prone to follow strictly the indications of the clinicians' respect to younger patients undergoing primary prevention. This potential flawing factor should be addressed and discussed. Please provide pre-treatment parameters that could permit us to understand the indication of statins: Previous AMI, Recent AMI, Primary prevention, hyperlipidemia, diabetes, etc.
The population selected to participate in this study where those with no previous cardiovascular disease. This fact was assessed excluding all subjects who declared a personal history of cardiovascular disease at the entrance of the cohort and based on the principal diagnosis assigned during hospitalization. As all patients are following primary prevention, we consider that this possible bias has been controlled. In relation to pre-treatment parameters, as all the subjects included are primary-prevention, we have added some information regarding their cardiovascular risk factors in table 1. In this table, information about smoking status, hyperlipidemia, glycemia, body mass index and CV Score are available.
No data are reported regarding the need to stop the treatment due to Adverse Events: how many had an increase of CPK, transaminases, or other justifiable cause to stop the treatment.
This is indeed an important aspect but, unfortunately, this information is not available. As this is a limitation of our study, we have added it in the discussion as follows.
“Also, we do not know the causes of no persistence, so they could be related with adverse events or a medical decision”.
How do the authors explain the lower mortality (numerical) of the non-persistent and the significantly higher time to hospitalization and MACE? Please answer and add also a sentence in the discussion.
It is true that, numerically, mortality and MACE are lower in non-persistent. But, as we explain in the discussion, these results are not statistically significant and the number is very low. So, it is necessary a longer follow-up to get a proper explanation.
As we have pointed out in the discussion, only 7 deaths due to CVD occurred in our study population, 4 in the persistent group and 3 in the nonpersistent group. For this reason, we could not analyse deaths by CVD. We expect that this limitation will change in the following years.
Regarding time to hospitalization, we have added the following sentence to the discussion section:
“The frequency of all-cause hospitalization, CV events, MACE, and death was similar for persistent and nonpersistent users. Regarding the time to hospitalization, it was significantly higher in non-persistent. As we are considering any cause of hospitalization, we cannot associate it to treatment.”
The comorbidity network represents one of the most engaging results of this study since the interrelation between risk factors represents one of the most intriguing and underinvestigated topics but unfortunately, Figures 1, 2, and 3 are not correctly uploaded and should be resubmitted.
Sorry for the inconvenience. We have uploaded figures 1, 2 and 3 again.
The number of co-occurring diagnoses was higher for nonpersistent than persistent users. Among nonpersistent users, there was a high probability of comorbid chronic kidney disease and hypertensive kidney disease. Do the authors think the non-persistence could be due to AE or to the excessive load of drugs (number of active drugs prescribed) in these patients?
Thank you very much for this comment. We have added this point to the discussion as follows:
“Comorbidities were more prevalent in nonpersistent patients. This fact could be explained as a consequence of non-persistence, but also as a cause, because patients with a high number of diseases tend to reduce drugs to the minimum necessary number. The strong association observed between chronic kidney disease and hypertensive kidney disease in non-persistent deserves further research, as it could be related to adverse effects or to other coprescriptions, which could be also the reason of non-persistence”.

Reviewer 3 Report
The study entitled ‘New users of lipid lowering drugs for primary 2 prevention of cardiovascular disease: the impact of 3 treatment persistence on morbimortality by Palacio and co-authors investigated a new users cohort of lipid-lowering drugs for primary prevention of cardiovascular disease (CVD) characterized all-cause and cardiovascular event-related morbidity and mortality, comorbidity networks and investigated the effect of treatment persistence on morbimortality. This study design is poor. It lacks clear objective, rationale, results, overall language is incomprehensible.
It is not clear why authors are calling this a follow-up study as it is running parallel with the AWHS study.
Abstract
Objective is poorly defined and highly confusing. Specifically, what authors meant by cardiovascular disease characterized all-cause morbidity and mortality?
Comorbidity networks? Which comorbid networks? What does author mean by this?
There are repetitive lines in the abstract. Line 29 is repetitive of line 25. Same information is provided both the lines.
In conclusion, authors have mentioned that persistence with lipid lowering drug has a protective effect against major adverse CV events (MACE) despite non-significant results. What is the basis of providing such conclusions?
Introduction
Introduction contains repetitive information which is quite distractive.
The authors should paraphrase the objective of the study, as it is not clear whether they want to investigate the effect of treatment persistence on CVD morbidity and mortality or treatment persistence itself is one of the end-point.
Methods
The authors have mentioned that their study is a follow up study of AWHS study. But they started monitoring patients’ soon after the subject enrollment was completed in the AWHS study. This does not make in sense. The connection between this study and AWHS study is lacking regarding subject selection. Study design is completely obscure.
The authors haven’t provided accurate information about the AWHS study such as:
- The final sample size of 5,678 workers is incorrect. In AWHS study it was 5,400.
- The objective of the AWHS study does not mentioned CV risk factors, instead factors associated with metabolic abnormality was included.
The authors have used teh term new users for the subjects. When there is no comparison with subjects with prior use of lipid lowering drug then what is the need of using the term ‘new users’. This is highly confusing.
Line72-23: What are the codes provided? No information is given in the manuscript?
Poorly-defined inclusion-exclusion criteria and end-points are not clear.
Line 78: what do analytical variables mean?
Line 99: What is the difference between CV events and major adverse CVD events?
Line 101-103: Why authors have chosen CV events and MACE as two end-points when the CVD codes for both are same?
It is not mentioned for how long the patient does has been taking lipid-lowering agent before performing any assessment?
Which lipid lowering drugs were taken by the subjects is not mentioned in the methods?
What comorbidities are parts of comorbid network analysis?
Results
No information is provided on patients who have discontinued the treatment or withdrew consent? This information is vital and is not included.
Line 158: What was the cause of death in the 33 subjects? Was it related to lipid lowering drug? What was age range of the subject who died?
In line 71, authors have mentioned that they excluded subjects diagnosed with any CVD before starting lipid-lowering treatment. Whereas, in line172, authors have mentioned that among CVD patients diagnosis of chronic ischemic heart disease and acute MI co-occurred. These statements are contradictory. Also, by showing association between acute MI with chronic ischemic heart disease and hypertensive chronic kidney disease with chronic kidney disease what does authors wants to convey or add to the current knowledge? It is an expected association.
Discussion
Extremely poorly written discussion.
Line 244: The authors have mentioned that they have included patients with dyslipidemia. This should be included in the method section.
Line 251-252: No where authors have mentioned that they also assessed air flow obstructions. Such information should be included in methods. Why they assessed this parameter? What is the significance?
What do authors mean by whole population?
Line 286-287: What is Farmasalud database? It should be mentioned in the methids.
Line300: what is P25, P75???
Author Response
On behalf of all the co-authors I would like to thank the reviewer for his/her valuable comments, which have been very useful to improve the quality of our manuscript. Please see our point-by-point response (cursive) to the comments below.
The study entitled ‘New users of lipid lowering drugs for primary 2 prevention of cardiovascular disease: the impact of 3 treatment persistence on morbimortality by Palacio and co-authors investigated a new users cohort of lipid-lowering drugs for primary prevention of cardiovascular disease (CVD) characterized all-cause and cardiovascular event-related morbidity and mortality, comorbidity networks and investigated the effect of treatment persistence on morbimortality. This study design is poor. It lacks clear objective, rationale, results, overall language is incomprehensible. It is not clear why authors are calling this a follow-up study as it is running parallel with the AWHS study.
Thank you for your comments. We have rewritten the objective, as well as the methods section, in order to clarify it. We have also made changes across the manuscript in order to improve its readability.
Regarding the study design, this work belongs to the AWHS study. We have rewritten this part of the methods section to clarify it.
Abstract
Objective is poorly defined and highly confusing. Specifically, what authors meant by cardiovascular disease characterized all-cause morbidity and mortality?
We have rewritten the objective as follows:
“The objective of this study was to analyse persistence to lipid-lowering drugs for primary prevention of cardiovascular disease (CVD) in a new users cohort, to explore all-cause and cardiovascular related morbidity, comorbidity and mortality in this group and, finally, to study the relationship between persistence and morbimortality”.
Comorbidity networks? Which comorbid networks? What does author mean by this?
Unfortunately, the low number of words available in the abstract do not allow us to explain comorbidity networks in this section. We have rewritten this sentence to make it clearer as follows:
“The association between comorbidities was explored using comorbidity network analysis.”
To complete this information, in the methods section we have explained what comorbidity networks are as follows:
“We next performed a comorbidity network analysis based on hospitalization diagnoses (CMBD)16. Comorbidity networks were generated to determine which diagnoses co-occur more frequently than expected by random chance. Nodes represent diagnoses and lines are the relationship between diagnoses. The magnitude between diagnoses indicates the number of times that two diagnoses co-occur in the sample, after discarding all pairs of diagnoses that only appear once. To quantify the "strength" of the association between two diagnoses, the number of patients with a given pair of diagnoses (observed cases) was divided by the number of individuals likely to have both diagnoses by chance (expected cases). Only positive associations were considered (having one diagnosis makes it more likely to have the other). A ratio >1indicates an “association” between diagnoses. To avoid inclusion of associations by chance proportion difference tests were performed: the resulting p-value indicates the probability that the difference between the observed and the expected findings is not exclusively due to chance. Based on these results we were able to select statistically positive significant associations (alpha=0.05). Finally, nodes that were not associated with any other node were discarded”.
There are repetitive lines in the abstract. Line 29 is repetitive of line 25. Same information is provided both the lines.
Thank you for this comment. Line 29 has been deleted.
In conclusion, authors have mentioned that persistence with lipid lowering drug has a protective effect against major adverse CV events (MACE) despite non-significant results. What is the basis of providing such conclusions?
Based on the results of multivariate analyses (both logistic regression and survival analyses) we observed that non-persistent users had a higher risk of suffering a MACE, but these results were not statistically significant. The number of words available in the abstract do not allow us to explain this aspect deeply in this section, but it can be observed in table 3 and figure 3. We have slightly rewritten this sentence in the abstract to clarify it:
“Adjusted analyses indicated a protective effect of treatment persistence, especially against major adverse cardiovascular events (MACE), but this effect was not statistically significant”.
Introduction
Introduction contains repetitive information which is quite distractive.
Introduction has been modified in order to improve it.
The authors should paraphrase the objective of the study, as it is not clear whether they want to investigate the effect of treatment persistence on CVD morbidity and mortality or treatment persistence itself is one of the end-point.
As suggested by the reviewer, we have paraphrased the objective of the study as it has been rewritten in the abstract, in order to make it clearer.
Methods
The authors have mentioned that their study is a follow up study of AWHS study. But they started monitoring patients’ soon after the subject enrollment was completed in the AWHS study. This does not make in sense. The connection between this study and AWHS study is lacking regarding subject selection. Study design is completely obscure.
Thank you for your comment. We have modified this paragraph as follows:
“This was a follow-up study, performed during the period 2010–2018. The population studied belong to the Aragon Workers’ Health Study (AWHS). The AWHS is a prospective longitudinal study of voluntary workers of a Spanish automobile assembly that was designed to evaluate the trajectories of traditional and emergent CVD risk factors and their association with the prevalence and progression of subclinical atherosclerosis in a middle-aged working population free of CVD at the beginning of the study. Recruitment of the AWHS cohort began in February 2009 and ended in December 2010. Further information on the AWHS can be found elsewhere12”.
The authors haven’t provided accurate information about the AWHS study such as: 1) the final sample size of 5,678 workers is incorrect. In AWHS study it was 5,400. 2)The objective of the AWHS study does not mentioned CV risk factors, instead factors associated with metabolic abnormality was included.
Thank for your appreciations. Although in Casasnovas et al. indicate that AWHS study included 5,400 people, this paper was written when recruitment had not finished. So, the final sample size was 5,678. Nonetheless, as we indicate in the methods section, we selected for this study only those workers who belong to AWHS cohort and began lipid-lowering treatment between January 1, 2010 and December 31, 2017. Analyses were restricted to new users (N=1,601). From those selected, we excluded individuals diagnosed with any CVD before beginning lipid-lowering treatment (N=113) and women women (N=64). So, the final study population was 1,424 men. We have modified this paragraph to make it clearer.
Regarding the objective of the AWHS study, it has been modified paraphrasing the study protocol as follows:
“The AWHS is a prospective longitudinal study of voluntary workers of a Spanish automobile assembly that was designed to evaluate the trajectories of traditional and emergent CVD risk factors and their association with the prevalence and progression of subclinical atherosclerosis in a middle-aged working population free of CVD at the beginning of the study”
The authors have used the term new users for the subjects. When there is no comparison with subjects with prior use of lipid lowering drug then what is the need of using the term ‘new users’. This is highly confusing.
New users are defined as those who had not received any lipid-lowering drug prescription during the 6 months preceding the start date. For this reason, the persistence with this medication was evaluated between the time of treatment initiation and discontinuation during a follow-up period of 1 year (starting July 1, 2010). This is the common way to calculate persistence.
Line72-23: What are the codes provided? No information is given in the manuscript?
Sorry for the inconvenience, but in line 73 of my manuscript´s version there are no codes.
Different classifications have been used in this manuscript. First one, the Anatomical Therapeutic Chemical [ATC] which is a classification system where the active substances are divided into different groups according to the organ or system on which they act and their therapeutic, pharmacological and chemical properties. There is a reference in the text regarding this classification:
WHO Collaborating Centre for Drug Statistics Methodology. WHOCC - ATC/DDD Index. [Online]. doi:10.1002/0471684228.egp13486
The other classification used is the International Classification of Diseases, Tenth Revision [ICD-10]. This is also a well-known classification to classify morbidity and mortality.
We hope that this information has answered your question.
Poorly-defined inclusion-exclusion criteria and end-points are not clear.
We have rewritten this part of the methods section in order to clarify this aspect.
Line 78: what do analytical variables mean?
Analytical variables are those obtained from laboratory tests. We consider that this is a commonly used term.
Line 99: What is the difference between CV events and major adverse CVD events? Why authors have chosen CV events and MACE as two end-points when the CVD codes for both are same?
MACE is defined as “major adverse cardiovascular event”. This is a frequently used health result in cardiovascular epidemiology. MACE includes only some specific CV diagnoses and its definition can differ. In this case, ICD-10 codes included were I21 and I60-I63 (myocardial infarction, stroke and nontraumatic subarachnoid hemorrhage).
On the contrary, CVD event is a more sensitive definition. It includes a larger number of ICD-10 codes: G45, G46, G81-G83, I20-I28, I46, I49.0, I50 and I60-I79.
In our opinion, the evaluation of two different end-points improve comparability and interpretation of our results.
It is not mentioned for how long the patient does has been taking lipid-lowering agent before performing any assessment?
This aspect is explained in methods section, in “analyses” part, as follows: “New users of lipid-lowering drugs for primary prevention of CVD were classified as persistent or nonpersistent. Persistence with this medication was evaluated between the time of treatment initiation and discontinuation during a follow-up period of 1 year”.
So, we classified patients as persistent or nonpersistent after one year of follow-up. In other studies, persistence is calculated as the number of days a patient remains treated, but this was not the objective of our study.
Which lipid lowering drugs were taken by the subjects is not mentioned in the methods?
Thank you for your comment. As we indicate in the methods section, we selected those subjects treated with lipid-lowering treatments (ATC code C10). ATC code 10 include the following drugs:
C10A Lipid modifying agents, plain
- 1C10AA HMG CoA reductase inhibitors
- 2C10AB Fibrates
- 3C10AC Bile acid sequestrants
- 4C10AD Nicotinic acid and derivatives
- 5C10AX Other lipid modifying agents
2C10B Lipid modifying agents, combinations
- 1C10BA HMG CoA reductase inhibitors in combination with other lipid modifying agents
- 2C10BX HMG CoA reductase inhibitors, other combinations
We have included this information in the text, in the methods section, as follows:
“ATC code C10 include lipid modifying agents plain (HMG CoA reductase inhibitors, fibrates, bile acid sequestransts, nicotinic acid and derivates and other lipid modifying agents) and HMG CoA reductase inhibitors in combination with other lipid modifying agents”.
What comorbidities are parts of comorbid network analysis?
As we explain in the methods section: “We next performed a comorbidity network analysis based on hospitalization diagnoses (CMBD) obtained”. So, all the diagnoses suffered for our patients during the follow-up were included in the comorbidity network.
In total, we obtained 447 different diagnoses, codified by ICD-10. The most frequent diagnoses was lipid metabolism disorders (E78), which appear 259 times, followed by primary hypertension (I10) which appear 205 times. This information has been added to the manuscript.
Results
No information is provided on patients who have discontinued the treatment or withdrew consent? This information is vital and is not included.
As it is stated in the manuscript, all participants signed informed consent to participate. Unfortunately, we have no information of those who withdrew consent. Nonetheless, 94.5% percent of workers consented to participate in the AWHS.
Regarding those patients who have discontinued the treatment, they have been included in our study as nonpersistent.
Line 158: What was the cause of death in the 33 subjects? Was it related to lipid lowering drug? What was age range of the subject who died?
As it was stated in the discussion, only 7 deaths occurred due to CVD in our study population. The most frequent cause of death was lung cancer (9 deaths). The second one was myocardial infarction (5 deaths). Two men died of liver cancer. Two deaths were by unknown cause. The median age of death was 60 years, range 47-67 years old, as it is a young cohort.
We have completed this information in the discussion section as follows:
“There were only 33 deaths in the cohort during the follow-up. The most frequent cause of death was lung cancer (9 deaths). Only 7 people died because of CVD in our study population, 4 in the persistent group and 3 in the nonpersistent group. For this reason, we could not analyse deaths by CVD”
In line 71, authors have mentioned that they excluded subjects diagnosed with any CVD before starting lipid-lowering treatment. Whereas, in line172, authors have mentioned that among CVD patients diagnosis of chronic ischemic heart disease and acute MI co-occurred. These statements are contradictory. Also, by showing association between acute MI with chronic ischemic heart disease and hypertensive chronic kidney disease with chronic kidney disease what does authors wants to convey or add to the current knowledge? It is an expected association.
Thank you for your comment. As it is stated in the results section, we considered only hospitalizations that occurred after starting lipid-lowering treatment. Nonetheless, it is true that this affirmation is confusing, so we have modified it as follows: “Red indicates diseases related to CVD or CV risk factors, such as diabetes or hypertension. All other diseases are shown in blue. Diagnoses of chronic ischemic heart disease and acute myocardial infarction co-occurred 17.6 times more often than expected by chance, and were also associated with long-term drug therapy”.
Regarding results, we agree that those presented are expected associations, but they are the strongest associations found. That is why we show them. In our opinion, the novel approach of our study is the analysis of the differences between persistent and nonpersistent subjects.
Discussion
Line 244: The authors have mentioned that they have included patients with dyslipidemia. This should be included in the method section.
Thank you for your comment. We have changed this sentence in the discussion because this was confusing. The new sentence is: “To the best of our knowledge, this is the first study to perform a comorbidity network analysis of patients treated with lipid-lowering drugs”.
Line 251-252: No where authors have mentioned that they also assessed air flow obstructions. Such information should be included in methods. Why they assessed this parameter? What is the significance?
Airflow obstructions are included in the comorbidity networks as one of the 447 diagnoses evaluated. This diagnosis corresponds to ICD-10 J44 code “Other chronic obstructive pulmonary disease”. It is a diagnosis assigned to the patient at hospital discharge by his/her doctor.
What do authors mean by whole population?
We have changed “whole population” to “for the entire group studied” to clarify it.
Line 286-287: What is Farmasalud database? It should be mentioned in the methids.
Thank you for your comment. Farmasalud database is explained in methods section as follows:
“Information on lipid-lowering treatments (ATC code C10) was obtained from the Medication Consumption Information System of Aragon (Farmasalud) for the period 2010–2018. This database stores information on drugs dispensed by pharmacies for prescriptions issued via the Aragon Health System. For each subject, we obtained information on the dispensing date, the ATC code of the dispensed drug, the number of defined daily doses (DDD), and the number of packages dispensed”.
Line300: what is P25, P75???
These abbreviations have been replaced by the complete term “percentile”.

Reviewer 4 Report
Comments for Authors
In this paper, the authors investigated the effect of treatment persistence on morbimortality in a group of patients (N=1,424 men) between January 1, 2010 and December 31, 2017. Only 38.7% of users were persistent with treatment. They found that nonpersistent users of lipid-lowering drugs have a higher number of co-occurring comorbidities than persistent users. The authors concluded that persistence with lipid-lowering drugs has a protective effect, especially against adverse cardiovascular events (MACE). However, more information and data are strongly required to add to the revised manuscript, which should offer more insights into the prevention of cardiovascular disease by lipid-lowering drugs. In addition, because the data from these patients have been used and analyzed for many publications, it needs to clarify whether the current results are overlapped with the other previously published or ongoing reports from the research group.
Major Comments:
- The entire treatment period for each patient should be defined between persistent and nonpersistent users during the period from January 1, 2010 to December 31, 2017. It is important to known whether the treatment duration also plays a key role in the prevention of MACE.
- Because genetic factors play a critical role in the development of cardiovascular disease, it is unclear whether there is a difference in family history of MACE, and whether this risk factor could influence the results.
- In addition, it is unclear whether nonpersistent users had more bad behaviors such as eating unhealthy food, less exercise, alcohol drinking, and sleeping time compared to persistent users.
- Moreover, it is unclear whether nonpersistent users had more underlying conditions such as insulin resistance, diabetes, obesity, nonalcoholic fatty liver disease, hypertension, kidney disease, the metabolic syndrome compared to persistent users.
- It is unclear whether hypercholesterolemia, hypertriglyceridemia, hypertension, and abnormal lipoprotein and glucose metabolism were under control between persistent and nonpersistent users during the period from January 1, 2010 to December 31, 2017.
- The dosages of lipid-lowering drugs have an impact on the morbimortality in the persistent users, as well as between persistent and nonpersistent users
- The discussion was redundant and ambiguous and did not show the limitation of the current study design. It did not explain clearly whether and how other risk factors such as genetics, unhealthy diets, less exercise, and underlying conditions may have impacts on the development of cardiovascular disease between persistent and nonpersistent users.
- Finally, some sentences were not written clearly so that this reviewer cannot follow what the authors wanted to say. These sentences need to be rewritten. In addition, there are some grammatic errors in the article. It is strongly suggested reducing the unnecessary uses of abbreviations in the manuscript because they led to difficult reading.
Author Response
On behalf of all the co-authors I would like to thank the reviewer for his/her valuable comments, which have been very useful to improve the quality of our manuscript. Please see our point-by-point response (cursive) to the comments below.
In this paper, the authors investigated the effect of treatment persistence on morbimortality in a group of patients (N=1,424 men) between January 1, 2010 and December 31, 2017. Only 38.7% of users were persistent with treatment. They found that nonpersistent users of lipid-lowering drugs have a higher number of co-occurring comorbidities than persistent users. The authors concluded that persistence with lipid-lowering drugs has a protective effect, especially against adverse cardiovascular events (MACE). However, more information and data are strongly required to add to the revised manuscript, which should offer more insights into the prevention of cardiovascular disease by lipid-lowering drugs. In addition, because the data from these patients have been used and analyzed for many publications, it needs to clarify whether the current results are overlapped with the other previously published or ongoing reports from the research group.
We have added the following sentence to the Methods section, in order to clarify this aspect: “This study has not been previously conducted and current results are not overlapped with other previously published or ongoing reports”.
- The entire treatment period for each patient should be defined between persistent and nonpersistent users during the period from January 1, 2010 to December 31, 2017. It is important to known whether the treatment duration also plays a key role in the prevention of MACE.
Thank you very much for your suggestion. Indeed, it would be interesting to evaluate treatment duration in MACE prevention. Unfortunately, in this study we have just evaluated one-year treatment to classify patients as persistent or nonpersistent, as a way to homogenise follow-up time. Nonetheless, this is an aspect that we want to evaluate in future research.
2. Because genetic factors play a critical role in the development of cardiovascular disease, it is unclear whether there is a difference in family history of MACE, and whether this risk factor could influence the results.
This information was not available for all the subjects analysed. That is the reason that it was not considered in our study. Just for your information, family history of hypertension, diabetes, myocardial infarction and stroke was collected for a subgroup of 700 subjects. In these subjects, these were the results obtained:
|
Overall (N=1424) |
Nonpersistent (N=873) |
Persistent (N=551) |
p |
|
|
|
|
|
|
|
|
Family history of hypertension |
70 (10.0%) |
51 (11.1%) |
19 (7.9%) |
0.264 |
|
|
|
|
|
|
|
Family history of diabetes |
65 (9.3%) |
43 (9.4%) |
22 (9.1%) |
0.999 |
|
|
|
|
|
|
|
Family history of myocardial infarction |
37 (5.3%) |
25 (5.5%) |
12 (5.0%) |
0.922 |
|
|
|
|
|
|
|
Family history of stroke |
13 (1.9%) |
6 (1.3%) |
7 (2.9%) |
0.151 |
So, as far as we know, no differences existed for this subgroup in cardiovascular family history. Nonetheless, we have added a sentence in limits section as follows:
“Information related with behaviours, such as diet or exercise, or genetics, was not available for all the subjects included. That is the reason why we could not consider it”.
3. In addition, it is unclear whether nonpersistent users had more bad behaviors such as eating unhealthy food, less exercise, alcohol drinking, and sleeping time compared to persistent users.
Thank you for your comment. This question is similar to the previous one. Although adding this information to our analyses would be of great interest, this was not available for all the subjects included in the study. That is the reason why we have not analysed it. The only information available for all the subjects was smoking (available in table 1). We have added this point in limits section as follows:
“Information related with behaviours, such as diet or exercise, or genetics, was not available for all the subjects included. That is the reason why we could not consider it”.
As we consider this aspect a key fact, we are already exploring it for a cohort subgroup in which this information is available.
4. Moreover, it is unclear whether nonpersistent users had more underlying conditions such as insulin resistance, diabetes, obesity, nonalcoholic fatty liver disease, hypertension, kidney disease, the metabolic syndrome compared to persistent users.
We presented in table 1 all the information available for all the subjects in our study at the beginning of the follow-up. In this table, information about smoking status, body mass index, hypertension, total cholesterol, LDL cholesterol, glycemia and SCORE is available. The objective of table 1 is to know the basal status of our population and the existence of differences between persistent and nonpersistent group at the beginning of the study. As we can see, no statistically significant differences were observed between groups, with the only exception of smoking (higher prevalence of current smokers in persistent group).
The other information proposed by the reviewer, such as nonalcoholic fatty liver disease or kidney disease, was not available for all the subjects analysed. That is the reason that it was not included.
5. It is unclear whether hypercholesterolemia, hypertriglyceridemia, hypertension, and abnormal lipoprotein and glucose metabolism were under control between persistent and nonpersistent users during the period from January 1, 2010 to December 31, 2017.
Thank you for your suggestion. As this is an important fact, we tried to analyse it, but it was not possible. Although subjects are offered a yearly analytic, there is a high percentage of subjects missing each year. Also, even in those subjects with yearly analytics available, it was not possible to find a pre-treatment and post-treatment analytic, in order to study treatment effectiveness. The same happened to hypertension. That is the reason why we only talk about persistent or nonpersistent subjects, instead of controlled patients.
6. The dosages of lipid-lowering drugs have an impact on the morbimortality in the persistent users, as well as between persistent and nonpersistent users
Thank you very much for your comment. We totally agree that dosages have an impact on morbimortality; also, the treatment regimen, in order to evaluate treatment intensity. Unfortunately, Farmasalud do not allow obtaining this complete information yet, but we expect to evaluate this aspect in the near future. We have added this fact to limits section as follows:
“Some information is not available in Farmasalud, such as dosages or treatment regimen, so we could not estimate treatment intensity”.
7. The discussion was redundant and ambiguous and did not show the limitation of the current study design. It did not explain clearly whether and how other risk factors such as genetics, unhealthy diets, less exercise, and underlying conditions may have impacts on the development of cardiovascular disease between persistent and nonpersistent users.
We have rewritten the limits section, adding these points. The current limits section is as follows:
“Our study has some limitations, mainly related to the data sources used. New users were classified as persistent or nonpersistent based on information available in the Farmasalud database. Farmasalud collects information on drugs dispensed through the public healthcare system. Although some dispensations may not be included, it is thought to cover approximately 98.5% of the population. Some information is not available in Farmasalud, such as treatment regimen, so we could not estimate treatment intensity. Also, our analysis is based on the assumption that drug purchasing equates to drug consumption, which is not necessarily true. Nonetheless, pharmacy records provide an easy and affordable way to estimate persistence in a real-world context29. Also, we do not know the causes of no persistence, so they could be related to adverse events or a medical decision. Information related with behaviours, such as diet or exercise, or genetics, was not available for all the subjects included. That is the reason why we could not consider it. Limitations associated with the use of comorbidity networks should also be considered. We selected all diagnoses (principal and secondary) at hospital discharge after beginning lipid-lowering treatment. Nonetheless, some of these diseases may have been established before users began treatment, as sometimes secondary diagnoses are used to collect relevant clinical antecedents. Because we cannot rule out the possibility that some of these conditions were already present when the patient began treatment, the observed association between treatment use and comorbidity should be interpreted with caution. Other limitations include the small sample size of the nonpersistent group, which complicated detection of statistically significant associations between diagnoses in this group: the low number of CV events identified; and the median duration of follow-up (2,136 days; Percentile 25, 1,498; Percentile 75, 2,505), which although long, was insufficient to detect statistically significant differences between groups. The lack of significant differences may also be associated with the young age of the new lipid-lowering drug users (median age, 52 years). There were only 33 deaths in the cohort during the follow-up. The most frequent cause of death was lung cancer (9 deaths). Only 7 people died because of CVD in our study population, 4 in the persistent group and 3 in the nonpersistent group. For this reason, we could not analyse deaths by CVD”
8. Finally, some sentences were not written clearly so that this reviewer cannot follow what the authors wanted to say. These sentences need to be rewritten. In addition, there are some grammatic errors in the article. It is strongly suggested reducing the unnecessary uses of abbreviations in the manuscript because they led to difficult reading
Thank you for your suggestion. We have eliminated some of the abbreviations in order to clarify the manuscript. We have also made some changes in the text to improve it and a native English speaker has reviewed it.

Round 2
Reviewer 2 Report
Dear Authors,
Thank you for the thorough and careful revision addressing all the reviewers' relieves.
My best Regards
Reviewer 3 Report
The authors have addressed my comments in the revised manuscript. I have no further comments.